# State of the Art of Cardiac Amyloidosis

**DOI:** 10.3390/biomedicines11041045

**Published:** 2023-03-28

**Authors:** Nabil Belfeki, Nouha Ghriss, Mehran Monchi, Cyrus Moini

**Affiliations:** 1Department of Internal Medicine and Clinical Immunology, Groupe Hospitalier Sud Ile de France, 77000 Melun, France; nouha.ghriss@ghsif.fr; 2Department of Intensive Care Medicine, Groupe Hospitalier Sud Ile de France, 77000 Melun, France; mehran.monchi@ghsif.fr; 3Department of Cardiology, Groupe Hospitalier Sud Ile de France, 77000 Melun, France; cyrus.moini@ghsif.fr

**Keywords:** cardiac amyloidosis, heart failure, transthyretin, immunoglobulin light chain, treatment

## Abstract

Cardiac amyloidosis is defined by extracellular deposition of misfolded proteins in the heart. The most frequent cases of cardiac amyloidosis are caused by transthyretin and light chain amyloidosis. This condition is underdiagnosed, and its incidence has been continuously rising in recent studies because of the aging of the population and the development of noninvasive multimodal diagnostic tools. Amyloid infiltration affects all cardiac tunics and causes heart failure with preserved ejection fraction, aortic stenosis, arrythmia, and conductive disorder. Innovative, specific therapeutic approaches have demonstrated an improvement in affected organs and the global survival of patients. This condition is no longer considered rare and incurable. Thus, better knowledge of the disease is mandatory. This review will provide a digest of the clinical signs and symptoms of cardiac amyloidosis, the diagnostic tools used to confirm the diagnosis, and current symptomatic and etiopathogenic management considerations according to guidelines and recommendations.

## 1. Introduction

Amyloidosis refers to a group of diseases resulting from deposits of altered proteins forming insoluble fibrils in different tissues and causing organ involvement. The term “amyloid” was first used by the Berlin pathologist Rudolph Virchow in 1854 to describe macroscopic tissue abnormalities seen by using iodine staining [1].

To date, we sum 42 proteinic precursors responsible for amyloidosis. One third of them lead to systemic amyloidosis [2].

The etiopathogenesis of amyloid deposits can be explained by three processes: excess protein production, abnormality or mutation of the native protein, and an intrinsic tendency to form amyloid. The elective deposits site and consequently clinical features depend on which “amyloidogenic” protein is misfolded. Hence, numerous organs can be involved, but cardiac localization is a major determinant of survival. Although traditionally considered a rare disease, recent advances in imaging and noninvasive diagnostic tools have led some to consider that cardiac amyloidosis (CA) is underdiagnosed and often unacknowledged as a cause of cardiac disease [3].

CA results in a wide array of symptoms caused by a restrictive cardiomyopathy, including heart failure, conduction disease, reduced quality of life, and death. This condition is categorized into two subtypes: transthyretin cardiac amyloidosis (ATTR-CA) and immunoglobulin light chain cardiac amyloidosis (AL-CA). This article will provide a digest of clinical signs and symptoms of CA, diagnostic tools used to confirm the diagnosis, and current symptomatic and etiopathogenic management considerations according to guidelines and recommendations.

## 2. Epidemiology

The first description of cardiac amyloidosis was made by Ferris in 1936 in a post mortem autopsy [4]. Since then, numerous emerging studies are suggesting CA. Maurizi et al. found that CA had a 9% prevalence among patients who had an initial diagnosis of hypertrophic cardiomyopathy, and it increased with age: from 1% at age 40–49 years to 26% in patients aged > 80 years [5]. Moreover, the prevalence of CA in post mortem studies was 10–25% in elderly individuals (age > 80 years) [6].

We believe that ATTR-CA is more frequent than previously appreciated. In fact, ATTR-CA is found in 16% of patients with degenerative aortic stenosis, and one in seven patients referred for transcatheter aortic valve replacement had occult CA [7]. Likewise, the UK amyloidosis registry described an increased incidence of Wild-type transthyretin amyloidosis from 3% in the period of 1987–2009 to 14% during 2010–2020 [8].

This highlights that this condition is common and should be included in the differential diagnosis of unexplained cardiopathy in elderly patients.

### 2.1. Transthyretin Cardiac Amyloidosis (ATTR-CA)

Transthyretin (TTR), previously called prealbumin, is mainly produced by the liver in a tetrameric circulating protein form and transports thyroid hormones and vitamin A. There are two types of ATTR amyloidosis: (i) variant transthyretin-related amyloidosis (ATTRv)—a familial disorder arising from the misfolding of a variant transthyretin; and (ii) wild-type transthyretin amyloidosis (ATTRwt)—a sporadic disorder due to the misfolding of wild-type transthyretin [3]. 

The mutation of the TTR gene (chromosome 18) or TTR protein senile misfolding leads to a tissue deposit of an insoluble monomeric transthyretin, including the heart. Its accumulation in the extracellular space impairs the diastolic function of the myocardium, which is observed as the restrictive cardiomyopathy pattern. Moreover, we observe an impairment of calcium channel transport and a cell metabolism imbalance leading to cellular edema that harms cardiomyocytes [9].

Currently, more than 130 different mutations causing ATTRv amyloidosis are reported. The clinical presentation depends on the type of mutation. However, the peripheral nervous system and the heart are commonly affected [10].

Patients with ATTRv are often diagnosed after the 6th decade of life. *Val122Ile*, present in 3.4% of the African Americans, is the most common mutation in the United States, followed by *Thr60 Ala*, which is found in individuals of Irish descent. In France, a recent national registry data showed that 8% of patients referred for hypertrophic cardiomyopathy are TTR mutated [11].

ATTRwt is primarily seen in elderly persons and mainly manifests itself with the phenotype of cardiomyopathy. It is typically diagnosed at 70–75 years of age, with a male predominance. The development of new diagnostic tools, notably nuclear scintigraphy, led to the diagnosis of ATTRwt CA more than expected. The prognosis is poor, with a median survival of 2–3 years in untreated patients [12].

It is important to know that age alone should not be used to differentiate ATTRv from ATTRwt. Identifying and differentiating the two subtypes is imperative, as it triggers genetic counseling and family screening.

### 2.2. Light Chain Cardiac Amyloidosis (AL-CA)

Immunoglobulin light chain (AL) amyloidosis is a subtype in which a plasma cell or B cell dyscrasia, such as multiple myeloma, B cell lymphoma, and Waldenstrom macroglobulinemia, produces abnormally misfolded free light chains (lambda more than kappa) that aggregate into insoluble amyloid fibrils and deposit in organs [13].

It is the most common type of cardiac amyloidosis with an estimated prevalence of 1.2/100,000/year and about 2000 to 3000 new cases every year in the United States [14].

Cardiac manifestation is the second most common presentation after renal involvement, and in more than 95% of cases it is associated with other system involvement. 

The exact mechanism of organ damage is unclear, but there is some evidence that direct light chain toxicity causes apoptosis in cardiomyocytes in addition to increasing filling pressure, making AL-CA a “toxic and infiltrative” cardiomyopathy [15].

An urgent management is mandatory because of the rapid risk of progression with a mortality up to 50% within 4–6 months and a median survival of less than 12 months. [16]

Grogan et al. have reported a 25–30% incidence of sudden death within 3 months of diagnosis [17].

## 3. Clinical Manifestations

The clinical presentation of cardiac amyloidosis is linked to the total or subtotal infiltration of the myocardium and/or a possible direct cytotoxicity due to certain precursors, especially the free light chains. The typical clinical manifestation is the picture of heart failure with preserved ejection fraction (HFpEF). Nevertheless, atrial fibrillation is often the first manifestation of CA. Moreover, the diagnosis of “hypertrophic cardiomyopathy” in elderly patients who do not have longstanding hypertension should prompt the search for an underlying CA [18].

Clinical signs are not specific and include lower limbs edema, jugular vein distension, liver enlargement and tenderness, pleural effusion, ascites, and shortness of breath. The reduced ventricular filling limits stroke volume and can progress to orthostasis, syncope, and low-output heart failure. Thus, conventional heart failure therapeutics are not well tolerated [19].

Cardiac amyloid infiltration concerns all the myocardial structures. i—Ventricular infiltration leads to walls thickening with biventricular symmetrical concentric hypertrophic cardiomyopathy and an impaired motion with an aspect of small ventricles and large atria called restrictive cardiomyopathy. ii—Bi atrial dilatation causes impaired reservoir function and increased stiffness, leading to supra-ventricular arrhythmia, intracavitary thrombi, and systemic emboli [20]. iii—Infiltration of the endocardium favors the occurrence of valvulopathies (mitral and tricuspid insufficiency, aortic stenosis). iv— Infiltration of the pericardium may result in the presence of mild effusion and exceptional tamponade. v—Vascular infiltration promotes ischemia. vi—Conductive tissue infiltration results in chronotropic incompetence and conductive disorders such as sino-atrial block, atrioventricular block, bundle branch block, and QT prolongation. vii—Autonomic nervous system involvement contributes to dysautonomic syndrome [9]. Amyloid cardiomyopathy is predated by extracardiac symptoms, which can be used as red flags for diagnosis. Unfortunately, the clinical symptoms of both forms of amyloidosis (AL-CA and ATTR-CA) are protean, and because of a misconception about CA, this contributes to its underdiagnosis and thus makes early diagnosis difficult. The most common extracardiac signs and symptoms of all forms of amyloidosis are listed as follows: proteinuria and/or nephrotic syndrome, small fiber neuropathy or polyneuropathy, hepatomegaly, skin bruising, macroglossia, bilateral carpal tunnel, spinal stenosis, and biceps rupture [21]. Figure 1 sums up the clinical picture of CA.

## 4. Diagnostics

Along with a detailed history and physical examination, the diagnostic evaluation of cardiac amyloidosis includes laboratory studies and cardiac imaging.

### 4.1. Electrocardiography (ECG)

A 12-lead ECG is abnormal in 90% of cases and classically shows low voltage in the QRS complex across the precordial leads. A discordance between lower voltage on the ECG and left ventricular hypertrophy on cardiac imaging is suggestive of cardiac amyloidosis [22].

Other abnormalities are reported: pseudo-ST elevation in 22%, no R-wave progression in precordial leads in 64%, and signs of atrial fibrillation and conduction disturbances such as prolonged of P wave duration, PR interval, or QRS complex, are also linked to myocardial amyloid deposits [23].

### 4.2. Echocardiography

Echocardiography is a central diagnostic imaging tool for cardiac amyloidosis using modern techniques such as deformation analyses by means of the automatic tracking of myocardial acoustic markers (speckle tracking echocardiography, STP). It shows an echogenic left ventricular wall thickening (>12 mm) with granular sparkling. Moreover, it may involve the right ventricle as well as the atria. A small pericardial effusion is often observed without signs of hemodynamic relevance. The left ventricular ejection fraction is often preserved. 

By evaluating the shortening of cardiomyocytes regionally and globally, global longitudinal strain is a non- invasive method to detect left ventricular function impairment. 

Although it is not a proof of CA, the phenomenon of apical spearing seems to be highly suggestive of the diagnosis and have a prognostic value. It consists of an impairment of longitudinal strain in the basal myocardium and conservation in the apical segments. 

Other abnormalities can be detected such as thickening of the valve apparatus, atrial enlargement, and intra cardiac thrombus [24].

### 4.3. Cardiac Magnetic Resonance Imaging

Cardiovascular magnetic resonance imaging is an important tool for the diagnosis of CA and for differentiating CA from other cardiomyopathies. However, it is costlier and less widely available than echocardiography. At an early stage, the deposits of amyloid proteins lead to an increase in the extracellular volume. It is identified in the late gadolinium enhancement (LGE) sequences of the left ventricle and atria, represented by generalized subendocardial LGE-foci. Cardiovascular magnetic resonance has proven to be a key diagnostic tool when the clinical and cardiac echography presentations are suggestive of CA [25].

The measurement of extracellular volume Is emerging as a potential measure to track the progression of CA over time and its response to therapy [26].

The T1 mapping sequence detects early infiltrative myocardial involvement. Hence, myocardial T1 mapping > 1048 milliseconds has a sensibility of 80% and a specificity of 83% to differentiate CA from other causes of hypertrophic cardiomyopathy [27].

In 3 to 7% of patients, a biopsy proved CA. Therefore, a normal CMR cannot allow the diagnosis of CA [28].

### 4.4. Radionuclide Imaging

^99^Technetium (Tc)-phosphate-radionuclide scintigraphy is currently considered the cornerstone diagnostic tool for ATTR cardiac amyloidosis. The tracers used are: 3,3-diphosphono-1,2, propanodicarbdicarboxylic acid (DPD); pyrophosphate (PYR), and methylenediphosphonic acid (MDP) [29].

It demonstrated a high affinity for myocardial amyloid depositions due to an unexplained mechanism. ATTR-CA has an increased affinity compared with AL-CA. Nowadays, simple visual scoring and a semi-quantitative visual indices are applied using the Perugini 4-point scale (0 = no cardiac and normal bone uptake, 1 = mild cardiac uptake less than bone uptake, 2 = moderate cardiac uptake and relatively equal bone uptake, and 3 = high cardiac uptake and only mild or absent bone uptake) [30].

In various cohorts of patients with heart failure and echocardiographic and/or cardiac magnetic resonance imaging findings suggestive of cardiac amyloidosis, cardiac scintigraphy with the grade 2 or 3 uptake of the radioisotope could confirm the diagnosis of ATTR-CA only when combined with blood and urine testing to exclude a monoclonal protein [31].

Consequently, different societal guidelines recommend the non-invasive diagnosis of ATTR-CA using ^99^Tc-phosphate radionuclide bone scintigraphy, while highlighting its use in the appropriate clinical context after ruling out AL-CA. Physicians should pay attention to the fact that about 10% of patients may have a high Perugini score (>2) in authentic cases of AL-CA [32].

Some authors mentioned false positive uptake in other conditions such as Danon disease (glycogen storage disease type IIb), hypertrophic cardiomyopathy, or hydroxychloroquine cardiac toxicity [33,34].

### 4.5. Laboratory Testing

Routine laboratory parameters are mandatory to screen for impairment of affected organs (e.g., kidney, liver, bone marrow). The assessment of cardiac parameters should be determined in patients with suspected amyloidosis. Brain natriuretic peptide (BNP), the N-terminal prohormone of BNP (NT-pro BNP), and troponin T (TnT) are the most commonly used biomarkers of cardiac involvement for the diagnosis of CA. Normal measurement of these tests excludes the presence of CA [35].

Besides this, cardiac biomarkers are included in prognostic scores for amyloidosis and are helpful for monitoring disease progression and therapeutic responses [18].

If AL amyloidosis is suspected, clonal dyscrasia must be ruled out by the following tests: serum free light chain assay; serum; and urine protein electrophoresis with immunofixation. The combination of serum protein electrophoresis with immunofixation, urine protein electrophoresis with immunofixation and quantification of serum free light chain has a sensitivity of 99% for identifying abnormal pro-amyloidotic precursor in AL amyloidosis [36].

When ATTR CA is confirmed, genetic counselling and testing are necessary to screen for the presence or absence of TTR mutations to differentiate between ATTRv and ATTRwt. According to the type of mutation, ATTR-CA may be diagnosed in older patients, with a median diagnosis age of 74 years in V122I-TTR mutated patients [12].

Thus, we suggest that genetic testing should be performed even in elderly patients. 

### 4.6. Biopsy

Although the examination of endomyocardial biopsies remains the gold standard for the diagnosis of CA with a sensitivity of 100%, multimodal noninvasive tools have declined this procedure [37].

Amyloid deposits are confirmed using thioflavin T, most commonly Congo red. On light microscopy, Congo red-stained amyloid deposits will appear red or salmon-pink. The confirmation test is the appearance of characteristic birefringence under cross polarized light when the amyloid deposits look apple green. The experts highlight that the term “amyloid fibril” should be used for any cross β-sheet fibril [38,39].

Identification of amyloid should be followed by classification of the amyloid fibril protein. Although the gold standard for defining the type of amyloid remains mass spectrometry, immunohistochemistry or immunoelectron microscopy are routinely employed for amyloid typing [40].

If there is evidence of systemic amyloidosis, biopsies should be taken from other organs that could be involved (e.g., skin, salivary glands, kidney). Figure 2 shows amyloid deposits on a lip biopsy using Congo red staining. 

Figure 3 illustrates a diagnostic algorithmic approach to the diagnosis of cardiac amyloidosis. 

## 5. Differential Diagnosis

The main differential diagnoses are other causes of myocardial hypertrophy. Sarcomeric hypertrophic cardiomyopathy is the most frequent differential diagnosis and can mimic ATTR CA. However, the absence of extracardiac clinical features, a normal voltage QRS complex across the precordial leads without conduction system disease on electrocardiography, and abnormalities of the mitral valve apparatus (specifically excess anterior leaflet length and anterior displacement of the papillary muscles on echocardiography) are suggestive of this hypothesis [41].

Other causes of infiltrative cardiomyopathy should be on the differential diagnosis. Fabry disease is a rare X-linked inherited lysosomal storage disease due to α-galactosidase A activity deficiency that leads to an accumulation of globotriasylceramide (Gb3) in affected tissues, including the heart. Main cardiac involvement consists of left ventricular hypertrophy, myocardial fibrosis, heart failure, and arrhythmias. The extra cardiac manifestations suggestive of the diagnosis of Fabry disease are cochleo-vestibular and ocular symptoms, ischemic stroke, and unexplained proteinuria [42].

Likewise, cardiac involvement is reported in about 5% of patients with systemic sarcoidosis. Typically, it presents with conduction abnormalities, ventricular arrythmia, and heart failure. Extracardiac manifestations with lymph node enlargement, respiratory symptoms with interstitial lung disease can be of great help in diagnosing this condition [43].

## 6. Cardiac and Specific Treatment of Cardiac Amyloidosis

The management and treatment of cardiac amyloidosis involves two areas: (i) management of the cardiac symptoms and cardiac related complications, and (ii) specific (disease modifying) treatment of the underlying disease to stop or delay further amyloid formation and deposition [18].

### 6.1. Management of Cardiac Symptoms

The therapeutic procedure encompasses different aspects, including the treatment of heart failure, arrhythmias (atrial fibrillation and ventricular arrhythmias), conduction disease, and aortic stenosis [44].

#### 6.1.1. Heart Failure

Volume overload symptoms can be challenging to treat in patients with CA. Diuretics are the cornerstone of treatment for controlling congestive symptoms. Monotherapy or combination therapy with loop diuretics and thiazide diuretics or mineralocorticoid receptor antagonists can be necessary to increase natriuresis [45].

Finerenone, a new selective nonsteroidal mineralocorticoid receptor antagonist, demonstrated improvement of heart failure outcome according to the FIGARO-DKD (Finerenone in Reducing Cardiovascular Mortality and Morbidity in Diabetic Kidney Disease) trial. There is no specific data concerning CA [46].

The effects of the sodium-glucose cotransporter 2 inhibitor (SGLT2i) on decongestion in acute heart failure was assessed compared with placebo, patients treated with empaglifozin demonstrated an early, effective, and sustained improvement of symptoms [47].

Encouraging results concerning the use of SGLT2i in heart failure in patients with ATTR-CA from small, real world case series have been published [48].

Further, larger reports are needed to assess the impact and safety of this drug in CA.

However, aggressive diuresis may aggravate low stroke volumes and cardiac output in CA and result in orthostatic hypotension and acute renal failure. For this reason, salt restriction is the first-line intervention for volume overload. Encourage patients to wear compression stockings to minimize hypotension.

Traditional heart failure treatment such as Angiotensin-converting enzyme inhibitors/Angiotensin II type 1-receptor blockers are not well tolerated and should be used with caution if needed because of risk of symptomatic hypotension. Sacubitril limits the degradation of amyloid fibrils due to the inhibition of neutral neuropeptidase, and thus it is contraindicated in CA [21].

Patients with CA develop a compensatory reflex to compensate for low stroke volume, but betablockers impair this mechanism and can harm cardiac output, which is very dependent on the chronotropic response. Likewise, non-dihydropyridine calcium-channel blockers should be avoided because of their negative inotropic action, inhibition of heart rate response, and hypotension [49].

#### 6.1.2. Arrhythmias

##### Atrial Fibrillation

Atrial wall dilatation and increased afterload predispose to atrial arrhythmia. It is a very common complication, particularly in ATTRwt CA patients, with an estimated prevalence of 70% [50].

Huntjens et al. investigated the prognostic utility of atrial longitudinal strain on echocardiography and showed its correlation to global death as an independent factor [51].

Rate control can be challenging because of a very narrow window for optimal heart rate control and the poor tolerance of beta blockers and calcium channel blockers. Contemporary data suggest that amiodarone is the best tolerated antiarrhythmic agent in patients with CA [44].

Catheter ablation has a limited role in the management of atrial fibrillation in this population because of the high risk of recurrence [52].

Patients with CA are at high risk of atrial clot formation and thromboembolic events, independently of documented atrial fibrillation. Therefore, the European Society of Cardiology Working Group on Myocardial and Pericardial Diseases recommends anticoagulation irrespective of traditional cardioembolic prediction scores (i.e., CHA2DS2-VASCc) and strongly considers screening all CA patients for intracardiac thrombus despite appropriate anticoagulation prior to elective cardioversion. Current guidelines recommend anticoagulation therapy in all patients with cardiac amyloidosis and atrial fibrillation [18].

Anticoagulation is warranted, but bleeding risk must be assessed, especially in AL amyloidosis patients with macroscopic digestive amyloidosis and/or acquired factor X deficiency (due to the adsorption of factor X to amyloid fibrils).

Moreover, Di Lisi et al. recommend anticoagulation therapy in sinus rhythm CA patients with a large atrium and low bleeding risk [53].

Both direct oral anticoagulants and vitamin K antagonists can be used to provide thromboembolic protection in CA and have comparable efficacy and bleeding rates [54].

##### Ventricular Arrythmia

Ventricular arrhythmias appear to be frequent in CA and carry prognostic implications. There are no guidelines, and current recommendations are based only on case reports and observational studies. Current recommendations suggest implantable cardioverter defibrillators only for secondary prevention, with a preference for subcutaneous over transvenous devices. Implantable cardioverter defibrillator therapy is not associated with a mortality benefit in patients with CA [18].

More prospective studies are needed to determine when an implantable cardioverter defibrillator would be indicated in CA patients [55].

#### 6.1.3. Conduction Disease

Conduction disease includes sinus node dysfunction, atrioventricular block, and intraventricular block. These conditions are frequent in CA patients and cause fatigue, dizziness, syncope, and sudden cardiac death. In 2018, the American College of Cardiology/American Heart Association/Heart Rhythm Society guidelines indicated permanent pacing in prolonged His-ventricular interval [56].

More recently, the European Society of Cardiology working group recommends considering cardiac resynchronization only in high-paced burden suspected cases [18].

Larger studies of device therapy in this population are still needed.

#### 6.1.4. Aortic Stenosis

Symptomatic aortic stenosis is found to be associated with ATTR-CA. In one study, the prevalence of cardiac amyloidosis in patients referred for transcatheter aortic valve replacement was 1 in 7 patients [7].

Patients with CA and aortic stenosis have higher mortality rate. This approach must be used with carefully selected patients. We recommend screening for amyloidosis for every patient with aortic stenosis. This strategy will allow for early, specific treatment [57].

Transcatheter aortic valve replacement is an interesting therapeutic approach in amyloid aortic stenosis. Experts recommends a special awareness during procedure because of risk of perioperative atrioventricular block [18].

#### 6.1.5. Orthostatic Hypotension

Orthostatic hypotension is due to an impaired release of catecholamines and is exacerbated by the low cardiac output. The management of this complication is difficult. All drugs that may affect the sympathetic nervous system, such as alpha blockers, must be discontinued [58].

Antidepressants are being reconsidered. Anemia should be treated. Compression stockings should be used to ameliorate venous return to the heart. Pharmacological approach consists in considering midodrine, an alpha-1 adrenoreceptor agonist, which increases vascular tone. Besides, droxidopa, an amino acid metabolized into norepinephrine, can be used as an adjunct to midodrine [58].

Last but not least, synthetic mineralocorticoids like fludrocortisone can be used to increase renal sodium and water retention [59].

### 6.2. Specific Treatment of Cardiac Amyloidosis

#### 6.2.1. Specific Treatment of ATTR Amyloidosis

The aim of the therapy is to reduce the production of mutated (liver transplantation) and overall TTR (genetic silencers) or stabilize circulating TTR molecule (stabilizers), preventing their dissociation or cleavage into amyloidogenic fragments. However, stabilization or silencing the expression of TTR protein seem to be the major therapeutic approaches.

Tafamidis stabilizes TTR tetramers and inhibits their dissociation into amyloidogenic monomers. It is indicated in ATTR cardiac patients with reasonable expected survival. Mauer et al. showed that Tafamidis reduced all-cause mortality by 30% and cardiovascular hospitalization by 32% versus placebo. In addition, quality of life, myocardial function, and cardiac biomarkers improved with Tafamidis [60].

Patirisan is a siRNA silencing TTR expression that has already been approved for the treatment of hereditary ATTR polyneuropathy stages I and II [61].

The European Society of Cardiology Working Group on Myocardial and Pericardial Diseases suggests that Patisiran could be considered in ATTRv patients with cardiac involvement in whom gene silencers are prescribed due to symptomatic neurological disease [18].

#### 6.2.2. Specific Treatment of AL Amyloidosis

For AL-CA, the aim of the treatment is to stop production of amyloidogenic light chains by monoclonal plasma cells. Based on recent published data in clinical trials, the treatment of choice consists of an association of monoclonal anti-CD38 antibody (daratumumab), an alkylating agent (cyclophosphamide), a proteasome inhibitor (bortezomib), and dexamethasone [62].

Depending on treatment response and eligibility for autologous stem cell transplantation, further therapeutic steps and regimens must be defined for the individual patient. Venetoclax, a Bcl-2 inhibitor agent, seems to be an interesting alternative in relapsing patients who are carriers of translocation t(11;14) [36].

## 7. Conclusions

Cardiac amyloidosis is an underestimated cause of infiltrative cardiomyopathy. Its incidence is rising for the aging demographic but also because of the development of non-invasive diagnostic tools. It is no longer considered a rare and untreatable disease. A compilation of workup consisting of imaging examinations (echocardiography, cardiovascular magnetic resonance imaging, and nuclear scintigraphy) and immunological assessment can lead to a precise diagnosis of AL-CA or ATTR-CA. Therefore, a methodical approach is mandatory for a rapid diagnosis and the initiation of etiopathogenic treatment as well as the symptomatic management of the cardiovascular manifestations.

## Figures and Tables

**Figure 1 biomedicines-11-01045-f001:**
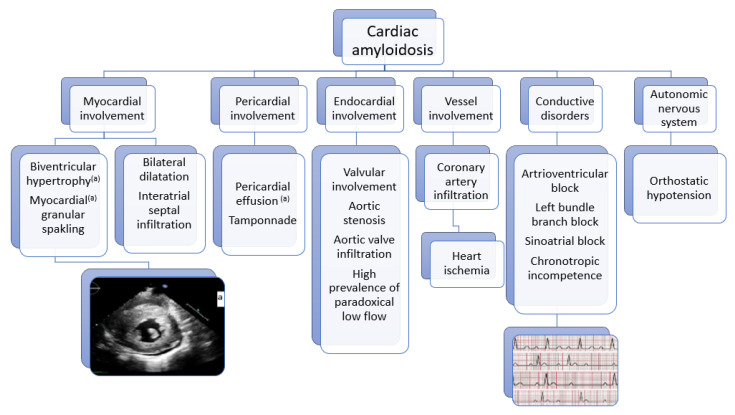
Clinical picture of CA.

**Figure 2 biomedicines-11-01045-f002:**
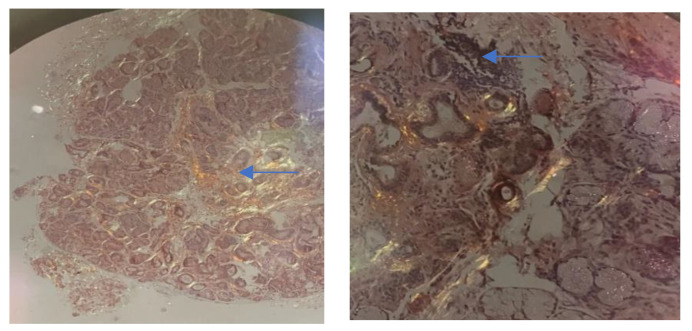
Amyloid deposits on a lip biopsy using Congo red staining. Blue arrow shows the amyloid deposits.

**Figure 3 biomedicines-11-01045-f003:**
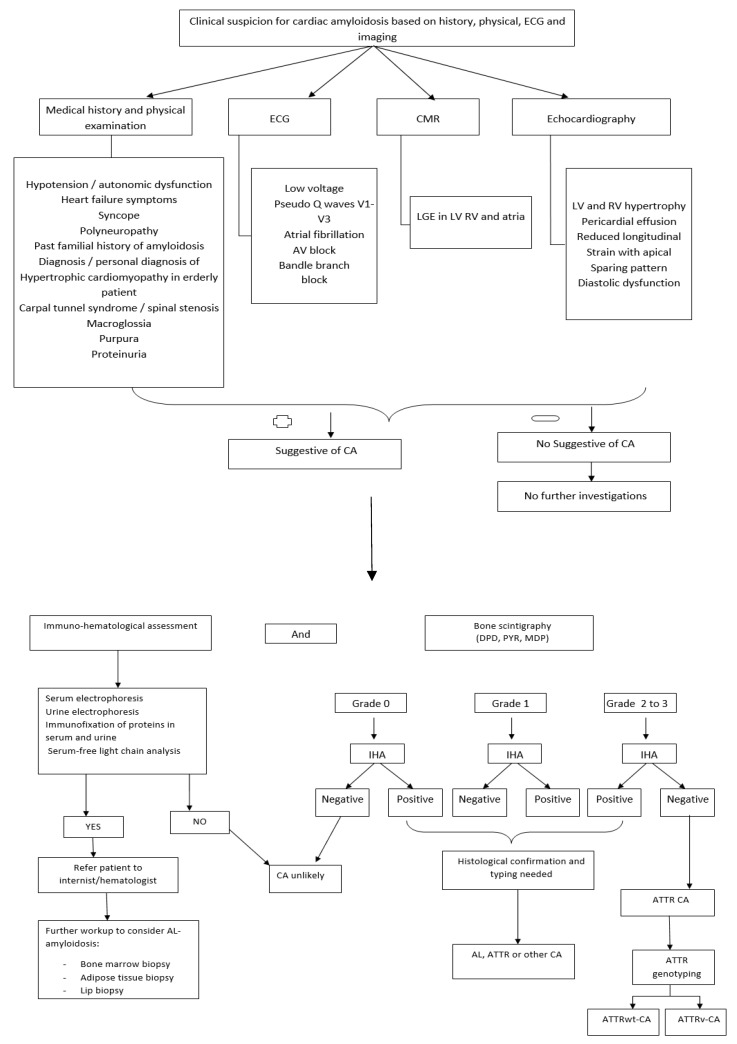
Diagnostic algorithmic approach to the diagnosis of cardiac amyloidosis. CA: cardiac amyloidosis; ECG: electrocardiography; CMR: cardiovascular magnetic resonance; AV: artrioventricular; IHA: Immuno hematological assessment; LGE: late gadolinium enhacement; DPD: propanodicarbdicarboxylic acid; PYR: pyrophosphate; MDP: methylenediphosphonic acid; LV: left ventricular; RV: right ventricular; AL: light chain amyloidosis; ATTR: transthyretin cardiac amyloidosis; ATTRwt-CA: wild-type transthyretin cardiac amyloidosis; ATTRv-CA: variant transthyretin-related cardiac amyloidosis.

## Data Availability

Not applicable.

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
