# Peer review of "State of the Art of Cardiac Amyloidosis"

_biomedicines, 2023, doi:10.3390/biomedicines11041045_

Round 1

Reviewer 1 Report

The review article titled "STATE OF THE ART OF CARDIAC AMYLOIDOSIS." by Belfeki et al. evaluates and discusses the current state of diagnosis and treatment of different forms of cardiac amyloidosis. Here are my recommended revisions.

Major concerns:

1.     The review article is severely lacking in citing references where needed, throughout the article, and the authors are advised to revise the entire manuscript and provide references for all information provided by citing the appropriate research articles. For instance (but not only), in sections 4.0 and 5.1, 5.1.3, the authors finish the entire section without citing a single reference. Please review this important problem.

2.     In section 1.1, The authors state, "Currently, more than 150 different mutations causing mt-ATTR amyloidosis are reported”. However, their reference article suggests 130 and offers no references for this number either. The authors are advised to find appropriate references to cite these numbers.

3.     In the second paragraph of the introduction, the authors state that there are 38 proteinic precursors of amyloidosis, for which they provide a reference from 2020. The authors are recommended to refer to the latest amyloidosis nomenclature to see the updated numbers (from Amyloid nomenclature 2022: update, novel proteins, and recommendations by the International Society of Amyloidosis (ISA) Nomenclature Committee).

4.     Also, in section 1.1 the authors refer to the hereditary ATTR as mt-ATTR. The authors are advised to review again the Amyloid nomenclature 2022: update, novel proteins, and recommendations by the International Society of Amyloidosis (ISA) Nomenclature Committee and follow the guidelines which suggest that “we recommend using the variant state after the name, such as ATTRwt, or ATTRv. Please note that v should be used for variant, not m (mutant) or h (hereditary), the latter since it may be mistaken for ‘human’.”

Minor concerns:

1.     The article's abstract can benefit from a description of the overview or scope of the review article in plain terms to ensure that the readers are aware of what to expect. For example, “This article provides a review of the management of patients with Cardiac amyloidosis including diagnosis, current treatment guidelines and recommendations …”

2.     The paragraph organization of the epidemiology section seems very distracting, and the authors are advised to reorganize it to ensure more clarity.

3.     Authors are advised to use African-American demographic rather than Afro-American.

Author Response

Dear Reviewer,

We are thankful for accepting reviewing our draft entitled STATE OF THE ART OF CARDIAC AMYLOIDOSIS submitted in biomedicines

Here are our answers according to your comments.

  • We added references in different parts of the manuscript.
  • We apologize for the mistake according to the number of ATTR mutations. Reference was added according to this data.
  • According to recent amyloid update, we sum 42 proteinic precursor responsible for amyloidosis. Modifications and recent references were added in the text.
  • According to the update on amyloidosis, we modified the nomenclature / abbreviations in the text.
  • We added text in the abstract according to reviewer’s comments.
  • The paragraph epidemiology was modified to make it easier for readers.
  • We used African-American rather than Afro-American according to reviewer’s comment.

Reviewer 2 Report

Title: STATE OF THE ART OF CARDIAC AMYLOIDOSIS

Type of manuscript: Systematic Review

Manuscript ID: biomedicines-2232321

Thank you for given me the opportunity to review this systemic review about "State of the art of cardiac amyloidosis".

There are many such review articles on cardiac amyloidosis and the added value is questionable. However, the article could have a special value if the authors succeed in giving it a special character, e.g., as a clinical guideline (but then also with clear recommendations).

Major Issues
1. The authors need to define the purpose of the article. Is the review intended to provide an overview of the topic of cardiac amyloidosis? Is it intended to be a guide to clinical practice or to raise general awareness of this disease? In either case, the goal should be emphasized in both the abstract and the introduction and should run throughout the text.

2. The authors state in the introduction: "This article emphasizes the necessity of an interdisciplinary approach to diagnose and manage cardiac amyloidosis according to the recent literature review." However, this fact is not further specified. What exactly is meant? Who should cooperate and how should the cooperation look like.

3. Many aspects are only briefly mentioned or touched upon, but not dealt with in depth. One is anxious to consult further literature because the article is insufficient. For example, anticoagulation: should patients receive anticoagulation even without atrial fibrillation? And if so, which therapy is more appropriate? Are there any studies or recommendations?

4. The diagnostic accuracy of the individual examination procedures should be better described (missing for ECG etc).

5. Measurement of extracellular volume (ECV) is essential for the diagnosis of CA using CMR. The authors should respond to this.

6. The section on differential diagnoses is roughly covered. Which are the most common and how can they be differentiated?

7. The conclusion is missing.

8. It should also address what is generally recommended when ICD and/or PM implantation becomes necessary. The authors should refer to current guidelines.

9. The text needs to be improved linguistically.

For example Page 4,5.1.2.1 AF: " Atrial wall dilatation and increased afterload predispose to atrial arrythmia."

Minor Issues
1. Abbreviations are mentioned once and are not used afterwards.

2. The disease patterns (ATTR-CA and AL-CA) are arranged under epidemiology, which is not a logical arrangement for me.

3. Figure 1 does not present clinical signs and symptoms, but rather the clinical picture?

4. Figure 3: If bone scintigraphy is negative with grades 2 and 3, is there really an ATTR-CA?

Author Response

Dear Reviewer,

We are thankful for accepting reviewing our draft entitled STATE OF THE ART OF CARDIAC AMYLOIDOSIS submitted in biomedicines.

Here are our answers according to your comments.

  • We add a purpose of our paper in the abstract and in the introduction. This review aims to report recent data on epidemiology and clinical manifestations on cardiac amyloidosis as well as recent published data on how we can diagnose this unrecognized condition. Moreover, we highlight the etiopathogenic and symptomatic treatment of this condition according to the recent recommendations.
  • There is a misunderstanding concerning the concept of multidisciplinary approach which mean the multimodal tools leading to definitive diagnosis of cardiac amyloidosis. Thus, we suppress this detail from the text.
  • We add more details according the management of cardiovascular manifestations of cardiac amyloidosis. We notice that there was a lack of references in different part of the text. Modifications were made according to the reviewer’s comments.
  • ECG exists in the text.
  • We added and detailed the place measurement of extracellular volume (ECV) for the diagnosis of CA using CMR.
  • We elaborate more text in the differential diagnosis chapiter. We highlight that sarcomeric cardiomyopathy is the most frequent differential diagnosis to recognize and detailed two other differential diagnosis (Fabry’s disease and cardiac sarcoidosis).
  • We wrote a conclusion.
  • ICD and/or PM implantation indications were specified in the text according to the recent published recommendations.
  • Linguistic modifications were performed.
  • We used abbreviations in the whole draft
  • We modified the title of figure 1 according to the reviewer comment.
  • As bone scintigraphy could be negative in some ATTRv mutations (tracer uptake depends on TTR fibril composition) and in rare subtypes of cardiac amyloidosis, we cannot completely rule out ATTR-CA in negative stage 2-3 bone scintigraphy.

Garcia-Pavia P, Rapezzi C, Adler Y, Arad M, Basso C, Brucato A, et al. Diagnosis and treatment of cardiac amyloidosis: a position statement of the ESC Working Group on Myocardial and Pericardial Diseases. Eur Heart J 2021;42:1554‑68.

Round 2

Reviewer 1 Report

The authors have addressed all concerns. 

Author Response

Dear reviewer,

Thank you for your comments. 

We hope that our draft is suitable for publication in its current form.

Kind regards,

N Belfeki

Reviewer 2 Report

Thank you for the revision of the manuscript, which has clearly benefited from it. In my view, further adjustments are not absolutely necessary.  

Author Response

Dear reviewer,

Thank you for your comments.

We hope that our manuscript is suitable for publication in its current version.

Kind regards,

N Belfeki MD